# The Relationship between Family Milieu and Music Education

**Tímea Szűcs**

Institute of Education Studies and Cultural Management, MTA-DE-Parent-Teacher Cooperation Research Group, University of Debrecen, 4032 Debrecen, Hungary; szucs.timea@arts.unideb.hu

**Abstract:** Our study aimed to map the family milieu of 21st-century music students. With the help of the applied social science methods, we studied what patterns facilitated children's music education and whether there were objectively justifiable differences in the socioeconomic status of music students and non-musical students. In our survey, we used a quantitative method in the form of paper-based, self-administered questionnaires. We sampled eighth-grade students of elementary art schools in several county seats that had a long history of teaching (n = 270) and eighth-grade students in several elementary schools (n = 285) as a control group. We assumed that students learning music are children of families with higher cultural capital, mostly with backgrounds in music education, who consider extracurricular activities investments. We tested our hypothesis using SPSS program, the methods included logistic regression and cluster analysis. Our results prove the existence of differences in the socioeconomic status of music and non-music students.

**Keywords:** music education; family milieu; socioeconomic status; cultural capital

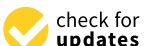



## 1. Introduction

The relationship between students' socioeconomic status, music learning and academic advancement is a remarkable research area in international education research literature (in the studies of Bresler 2002; Catterall et al. 1999; Miksza 2007; Hargreaves et al. 2003; Wen et al. 2020). These studies confirm that in their school achievement there are significant differences between students who are involved in musical activities and their peers who are not, with the obvious advantage leaning toward music learners. Moreover, music students of lower social status perform better than could be expected of those from their social position.

Earlier research studies have linked this to several skills and abilities such as efficiency, health, well-being and resiliency (Custodero 2002; Gick 2011; Portowitz et al. 2009; Szűcs 2019). In our study, we consider learning music as an early investment in the child's cultural capital.

Throughout the course of our research, we aimed to explore the socioeconomic background of children learning music, to find any differences between them and their non-musical peers. Meanwhile, we studied what factors influence their participation in music education. In summary, we sought to answer the question of whether learning music has an elitist function, (Andor 2002) a function compensating for disadvantage (as stated by Bácskai et al. (1972)), or both.

The study's novelty lies in the fact that in the review of the special literature we rethought the types of capital (Bourdieu 1999) using an art education perspective, that we might receive an answer to the following question: what is the social significance of music education? Besides this, we coined the term, *music education biography.* From among the methodological novelties, we were about to highlight the clusters of family milieu formed by the capital types.

We selected Borsod-Abaúj-Zemplén, Szabolcs-Szatmár-Bereg and Hajdú-Bihar counties as the location for our study since in these three border counties (excluding the central region) the number of elementary art schools is the highest.

When selecting the elementary art schools, we endeavoured to collect comparable data. Therefore, in all three counties we opted for elementary art schools in county seats with a long history of teaching music. The other reason for choosing these types of institutions was that depending on the school hosting the music lessons, different attitudes and motivations were required of students and parents alike. Taking music lessons elsewhere requires extra time, energy, and endurance.

The construction of the study begins with review of the special literature, followed by the methods section, and the results, discussion, and conclusions chapters.

## 2. Literature Review

In our study, we approach the theories on capital from an art education stance, seeking out possible points of correlation. The concept of capital, which had earlier been used in economics, became widespread in social sciences in the 19th century (Pusztai 2009). Within social sciences, capital can be defined as: a social benefit or advantage for the individuals, which determines their social position (Angelusz et al. 2006). Education Sociology research places great importance on these different types of capital when explaining the diversity of individual academic careers (Pusztai 2009).

Bourdieu's (1986) cultural reproduction theory distinguishes three types of capital: economic capital, social capital, and cultural capital. Economic capital refers to investible money and/or wealth. Social capital—or in other words capital of relationships—represents the interpersonal social network of the individual. Cultural capital speaks of their level of education. The different types of capital intensify each other and can be converted one into another. For instance, social capital can be transformed into economic capital.

Being published almost parallel to Bourdieu's study on forms of capital, Coleman (1988) almost wrote in detail about theories of capital. It is however important to note, though we do meet with three capital types in Coleman's theories (physical or objective capital, human capital, social capital), these types appear at almost the same time as Bourdieu's, and these, like Bourdieu's, can also be transformed, the approaches these authors have taken toward the topic are two different matters. In the case of Bourdieu's economic approach, we see the processes necessary for an individual to maintain his social status. The foundation of Coleman's (1988) theory is intentional action, which aims at satisfying needs at the individual level, the realization of which is affected by the environment. Of the capital types mentioned by both authors, we, in our study, focus more intently and in greater detail on social and cultural capital.

Social capital, one of the forms of capital, first appeared in the works of Hanifan, who first used the term in connection with the correlation between belonging to a community and school achievement. The author referred to the relations among people living in the same place, to mutual comradery, friendship, and goodwill (Hanifan 1916). Social capital, also according to Coleman (1990), is the intellectual assets gathered by mankind. We examined how Bourdieu and Coleman compare to one another, and we concluded that both Bourdieu and Coleman consider the functional value of social capital fundamental. Bourdieu believes that the individual's credibility or level of recognition is the benefit of relationships (authority capital, capital of obligations), whereas Coleman stresses the importance of the cohesion of the social network. This simply means that characteristics of the network such as its stability, closeness, density and attitude toward norms are important (Pusztai 2015). When examining the quality of relationships, the literature highlights strong and weak ties. Strong ties denote tight contact, a closed structure, and acceptance of common values. Weak ties indicate loose contact over significant spatial and social distances (Granovetter 1991). Putnam (1995) further developed his social capital theory along Coleman lines. Norms and trust are at the centre of Putnam's (1995) definition of social capital. Social capital is an investment, which aims to inspire individuals to achieve common goals and form group identity. In his opinion, social capital is based on a variety of relationship networks: trust, solidarity, social co-operation, the flow of information, the norm of mutual assistance and community development (Orbán and Szántó 2005).

He distinguished two types of relationships: one is bonding social capital, referring to close relationships (e.g., family), the other one is bridging social capital pertaining to more distant ones (e.g., acquaintances).

According to Dewey (1915), social capital is the intellectual assets gathered by mankind. In terms of building relationships, he too highlighted the importance of comradery and cooperation developed during shared schoolwork, which helps bridge social divides (Pusztai 2009). Putnam spoke about in-school and out-of-school social capital as well. In-school capital touches on the relationship between teachers and school management, on parents' involvement in decisions, and informal learning from peers. In contrast, social capital outside of school points to the strength of students' community and family relationships (Putnam 1995). According to Coleman, these parental networks or ties can support students if parents have similar ideas about the child's school career. Blau believes that the exchange of information among parents about parenting can be beneficial to the attainment of social status (Meier 1999).

German school climate research studies have proved that school rules, customs, social relationships, practice, and habits have a remarkable impact on the development of the students. According to Fend (1977), the school-specific norms, values, expectations, and the climate significantly influence the students' self-confidence and their view of success. The spirit of the school (Kozéki 1991) includes the moral concept, along with a system of values and norms. All these together are called the "learning environment" or *milieu* (Hradil 1995).

According to Bourdieu (1999) cultural capital exists in three forms: incorporated, that is, internalized; objectified (cultural goods) and institutionalized (qualifications). It takes time to accumulate capital, both in the case of the objectified and the acquired forms.

In Bourdieu's theory, school success is the result of the family's cultural capital investment, which can be manifested in different educational qualifications, cultural goods, erudition, and sophisticated behaviour (Pusztai 2009). The level of investment in cultural capital depends on how the different social classes relate to the future, to work, and to school (Bourdieu 1978). This way they enter the world of education and work with different chances. He believed that the cultural resources acquired in the family enabled the elite to pass favourable positions on to their children. This way of thinking has a stronger influence on the social position than the level of education or occupation (Róbert 1987). However, school also plays an important role in preserving social differences. In his research study, DiMaggio (1998) concluded that cultural capital has a significantly positive impact on school grades, however, it does not only promote the transmission of benefits from one generation to the next but also improves the chances of the lower strata of society. The theory of cultural mobility states that children from lower social classes can better utilize the cultural resources at their disposal than those of the elite. According to special literature regarding social mobility, it is the students' determined ambition that helps or hinders them in their school career. This is influenced, first and foremost, by their position in the social structure, which determines how much cultural and economic capital they possess. Secondly, it is defined by the students' micro-social network of relationships (Ainsworth 2002).

According to Ferge (1980), who examined the reproductive effects of cultural capital, the social heterogeneity of students evens out in schools attended in greater ratios by children of intellectual parents. So, the children of parents with higher levels of education influence the further-study intentions of children with lower socio-economic status. In this way, social mobility can become a reality.

The test results of Vitányi and colleagues showed that the class type was more definitive than socio-economic status, since music education also had an effect on social progression. The youth who came from poorer families and attended music primary school, all finished studies four years later that, regarding social hierarchy and standard of living, would allow them to continue their future lives at a higher level (Bácskai et al. 1972).

Blaskó (2002) studied whether the cultural reproduction theory or the cultural mobility theory proved to be working in Hungary. Her results showed that up until the 1960s cultural mobility was dominant, while cultural reproduction led in the 1970s.

The connections between parent training and social class reproduction are supported by international special literature. Extracurricular activities offer abundant opportunities to the children of middle-class parents, who responsibly try to maintain their social status. This responsibility fuels the market, provokes the state, and drives societal competition (Vincent and Ball 2007). Parents held extracurricular activities as a tool for learning vital skills and characteristics, which ensure their children's future professional and personal success. It is an interesting experience that during the course of the examinations difference was detected in the attitude of parents from Rome and Los Angeles. For Roman parents, the free-time and elective natures of these activities were important. To them, these were tools to help teach their children and to prepare them for adult life. In contrast, Los Angeles parents emphasized the need for their children to be committed and driven to perform successfully (Kremer-Sadlik et al. 2010).

After a brief historical overview, we would like to highlight how capital theories could be utilized in the context of music education. Students' extracurricular activities (music participation, sport) often seem to be an effective investment in both cultural and social capital (Kennedy 2002). Some researchers believe that extracurricular activities help disadvantaged students overcome challenges (Pusztai 2009).

Pusztai (2009) groups extracurricular activities into several categories. The first group comprises activities to counterbalance learning deficiencies or to acquire extra knowledge. The second group includes sports activities, the third covers activities that aid music literacy (learning music and participating in a choir). Participation in the public life of the school is the fourth, while the fifth group contains charity and religious activities. These extracurricular activities can influence school performance in many different ways. The impact of sport and art activities is manifested in the improvement of grades and successful entrance exams (Ho et al. 2003; Schellenberg 2004; Schmithorst and Holland 2004; Wong and Perrachione 2007). Extracurricular participation in the school's public life intensifies students' commitment to their alma mater, which has a positive impact on their academic achievement (McNeal 1999; Meier 1999). It has also been revealed that hyper-networked students (students with a dense network), engaging in multiple out of school activities perform well at school (Broh 2002). This phenomenon can be explained by their effective time management, endurance, good organizational skills, and a positive attitude towards work.

If the extracurricular activity does not take place on school premises, it requires investment, financial resources, time and determination from the parents. Therefore, it can be assumed that these activities are utilized by families with better financial standing or by middle-class families using out of the ordinary investment strategies (Pusztai 2009).

Members of the middle class are especially willing to invest in the future of their children (Vincent and Ball 2007), that they might maintain or raise their social status, even at the expense of having to give up something (e.g., holidays).

In Kovács' (2014) study, founded upon Bourdieu's theory, the child's social status was built upon the following social background variables: economic capital, cultural capital, subjective and objective financial situation, the level of education of the parents as well as the type of home settlement. In his opinion, the economic and cultural capital of the family determines how the family spends their spare time. More highly educated people more frequently visit theatrical performances and concerts and are more avid readers of more sophisticated books. The consumption of cultural goods is influenced by financial resources, though, cultural capital also plays an important role (Pusztai 2009). However, multitudes of studies show other results. In even more European countries and nations outside of the EU, research was carried out, which examined the definitive nature of social class and cultural capital. The nature of these two elements determines school results and social status. Based on the outcome of those studies, the importance of school orientation and cultural capital to parents varied from place to place (Dumais 2005b). Aside from this, more researchers found that, in American schools, cultural capital is not only beneficial to

students with good, advantageous backgrounds, but also good for every student who has this. It is on this phenomenon that the "cultural mobility" model is based (DiMaggio 1982).

Participation in music and other arts builds the cultural capital of children since theoretical studies (solfege, music history, music literature, music theory) and attendance of events broaden their level of cultivation (Janurik 2020). Bourdieu (1999) also believes that accumulating capital requires time. Learning music is a long process that requires regular practice. Therefore, if one aims to play a musical instrument even on a basic level, it takes several years of learning. The regularity and consistency demanded by music training also play an important part in the adult life of a child.

Continuing on, Bourdieu (1986) considers social capital a resource that is based on belonging to the same group. Building social networks is crucial for the individual as this way they can strengthen their social status. To begin, in elementary art schools,[1] there is a strong sense of belonging to a group. Students with similar interests attend these schools, where singing together in a choir or making music in a band strengthens these bonds. These relationships live on outside the school, lasting even into their adulthood. These lasting relationships can significantly contribute to the growth of their social capital.

Coleman's social capital theory (Coleman 1988) links playing music to its personality developing and community features (Sichivitsa 2007). Community music activities add to social networks, in this way increasing social capital, thus helping academic achievement.

In playing music, trust is key, between the teacher and student, and between them and the others involved in the musical production. At a concert trust appears when we have full confidence in the accompanying pianist's or singer's ability to join in the music at the right time, the correct pitch and right tempo. Coleman (1988), Granovetter (1991) and Putnam (1993) all analyze social capital and social networks referring to this kind of trust.

Another important theory to cover is that of Bandura (1989). His social learning theory surmises that the environmental and social conditions in which the child's socialization is realized determine his or her future career and decisions. Furthermore, behavioural patterns of family members, friends and peers involved as role models in the socialization process also influence the child's attitude. Thus, in families where there is someone who plays music, a positive attitude towards music can be developed, increasing the likelihood that, that child will want to learn music.

In the empirical portion of our study, in order to define cultural capital, we asked about parents' highest academic achievement, their musical biographies, the shared family habits of singing and playing an instrument, the size of the family library, the freetime habits of children, extracurricular activities, and the time devoted to such activity.

## 3. Methods

In our study, we hypothesize that music students are children of families with higher cultural capital, mostly with music education backgrounds, who consider different extracurricular activities investments, which support their advancement in the social hierarchy or help them retain their present status.

Our study approached the question of music education biography (previous musical experience or career path) from two sides—(1) from the parents' perspective, and (2) from the children's activities related to learning music. In connection with the music education path of the parents, we took into consideration their institutional music participation, that is, their music school studies in their childhood, as well as their current activities related to music. The latter includes the parents' skills to play a musical instrument with a distinction between taught or self-taught skills. Activities attached to choral singing, folk music playing and folk dancing were also included in this category. The students' music education biography also consists of two components: their parents' active involvement in their music education; for instance, singing or playing music together at home, and formal music education. Overall, music education biography was built up of these four components (Table 1).

**Table 1.** The biographical components of music education (Composed by author).

|  | Formal music education | Childhood music school studies |
|---|---|---|
| Parents' music education biography | The current musical activity of parents | • musical instrument proficiency<br>   ○ Learned from teacher<br>   ○ Self-taught<br>• singing, choir<br>• folk music, folk dance |
| Children's music education biography | Formal music education | Elementary art school |
|  | Parental participation in child's musical education | • singing together at home<br>• playing instruments together at home |

The analysis of the empirical study was based on our self-composed and self-recorded database titled "Learning music in Hungary 2017", which comprises data from paper-based, self-administered questionnaires for music students and the control group. The first three groups of questions contained queries to determine the social status of music learners (basic demographic variables, economic capital, social capital, and cultural capital). The other three groups of questions in the questionnaire were related to learning (school performance, plans for further studies and motivation for learning music).

We used the database of 8th-year students (13–14 years of age) from primary schools at ISCED level 2[2] at the time of the survey. The selection of age group was based on expert opinion. The reason for this is twofold: (1) the children have several years of experience in elementary art school, and (2) the tendency for teenage children to care about their parents' opinions (they do not attend school at the wishes of their parents, as is typical of younger generations, but it is of their own free choice). In earlier research in this field, during the course of a survey on the popularity of music classes, the ISCED age group 2 was also the target group (Janurik 2008; Janurik et al. 2021). All students learning music in the given age group attending the selected elementary art schools completed the questionnaire (N = 5207 persons; n = 269 persons). Regarding the control group, it was important that children with similar socio-economic backgrounds be allowed into the sample, just like learners of music, who had the opportunity to learn music, but they had chosen something else. Thus, these two groups are able to be analysed. Based on the student surveys from the Hungarian National Assessment of Basic Competences,[3] we chose the schools where neither "elite", nor "disadvantaged" students are represented, but rather schools that represent the middle field, the average.

During our study, SPSS was used for data analysis. That we might know how much family environment impacts the start of music education, logistical regression was carried out. The dependent variable of the analysis was whether students were learning music, the explanatory variables being components of the economic, social, and cultural capital. Regarding the economic capital, we examined both the objective and the relative financial situations. In the case of the objective financial situation variable, we took into account whether the family possessed real estate or not. Regarding the relative financial situation, we identified above average, average or sub average statuses.[4] Regarding social capital, when analysing the family structure, the traditional or non-traditional/modern family composition was determinant. When it came to having siblings, it was crucial to have at least two. Those respondents who followed the teaching of their church or who were religious in their own way were regarded as religious. When coding the components of cultural capital and looking at the highest level of education, the relevant factor was the fact that both the father and the mother had completed higher education. In connection with the size of the family library, those having more than 20 shelves of books were taken into consideration. Regarding the variables of singing or playing music together at home, whether these occasions happened or not was determinant. In the case of cultural

consumption habits, high consumers of culture[5] or those interested in culture were counted, but we did consider the stay-at-home types as well.

When analysing the families along with the different types of capital, key cornerstone similarities and differences were outlined between the family milieu of music students and that of non-musical pupils. By performing cluster analysis, we aimed to find groups showing similarities in the examined aspects and to describe the social status of music students more precisely. The categorization variables used for the analysis were the number of siblings, the parents' highest level of education, the religious practice of the children, their consumption of culture, the objective status of the material assets in the possession of the families and the parents' active participation in their children's music education. During the analysis six groups became clear to see: (1) academic milieu with traditional values; (2) academic milieu with high cultural capital; (3) academic milieu with low cultural capital; (4) less educated families with traditional values; (5) less educated families open to culture; (6) less educated families with low cultural capital.

## 4. Results

The results of the logistical regression on the role of family milieu in beginning music education can be seen below in Table 2.

**Table 2.** The effect of capital types on learning music. (Source: "Learning music in Hungary 2017" study, composed by author.).

|  |  | **Sig.** | **Exp (B)** |
|---|---|---|---|
| Economic capital | objective financial situation | 0.017 | 2.214 |
|  | relative financial situation | 0.005 | 0.254 |
| Social capital | religion | 0.008 | 2.418 |
|  | family composition | 0.048 | 2.157 |
|  | number of siblings | 0.587 | 1.203 |
| Institutionalized cultural capital | mother with a higher education degree | 0.001 | 3.48 |
|  | father with a higher education degree | 0.129 | 1.839 |
| Objectified cultural capital | size of the family library | 0.003 | 4.275 |
| Incorporated cultural capital | singing at home | 0.029 | 2.23 |
|  | playing music at home | 0.377 | 1.444 |
|  | cultural consumption habits | 0.000 | 10.658 |

From the analyed variables, all were significant coefficients except for the number of siblings, the father's degree and playing music at home. The degree of the mother, the size of the family library and culture consuming habits were considered most weighty or the strongest variables. The only negative impact was detected in the relative financial situation, meaning all the other factors positively influence the outset of learning music.

In finishing the cluster analysis along the examined views, we aimed to find groups that showed similarity. During the analysis, six clusters (1. academic milieu with traditional values; 2. academic milieu with high cultural capital; 3. academic milieu with low cultural capital; 4. less educated families with traditional values; 5. less educated families open to culture; 6. less educated families with low cultural capital) appeared.

In the entire sample, the cluster ratios have balanced out, as can be seen in Figure 1.

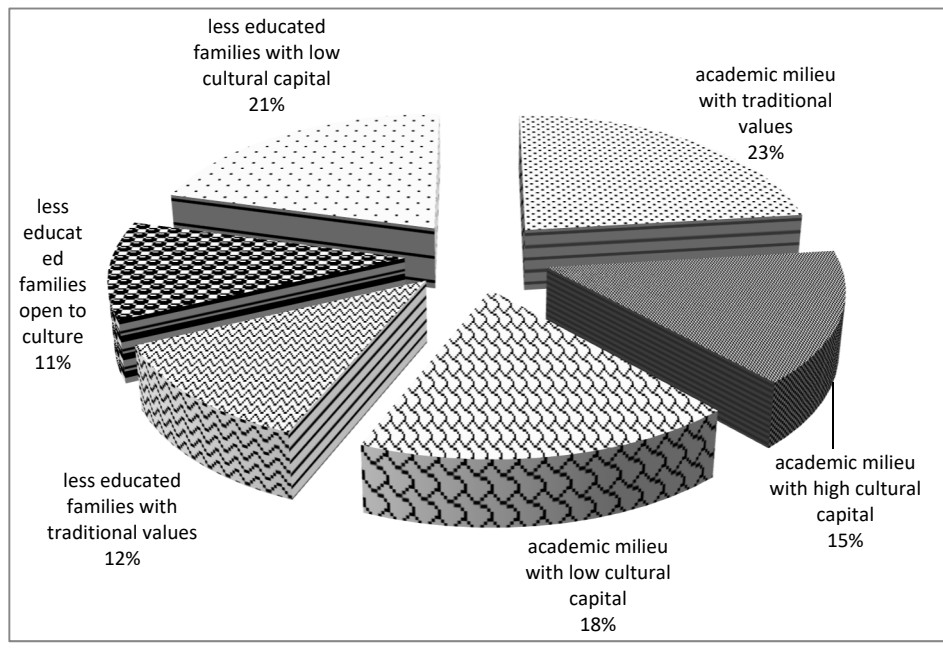

**Figure 1.** The family milieu clusters along with the different types of capital. (Source: "Learning music in Hungary 2017" study, composed by author).

Since discovering the characteristics of the music students and the control group lay at the center of our examination, we separately examined the cluster ratios of the two groups, and sharp contrasts came of it, which can be seen below in Figures 2 and 3.

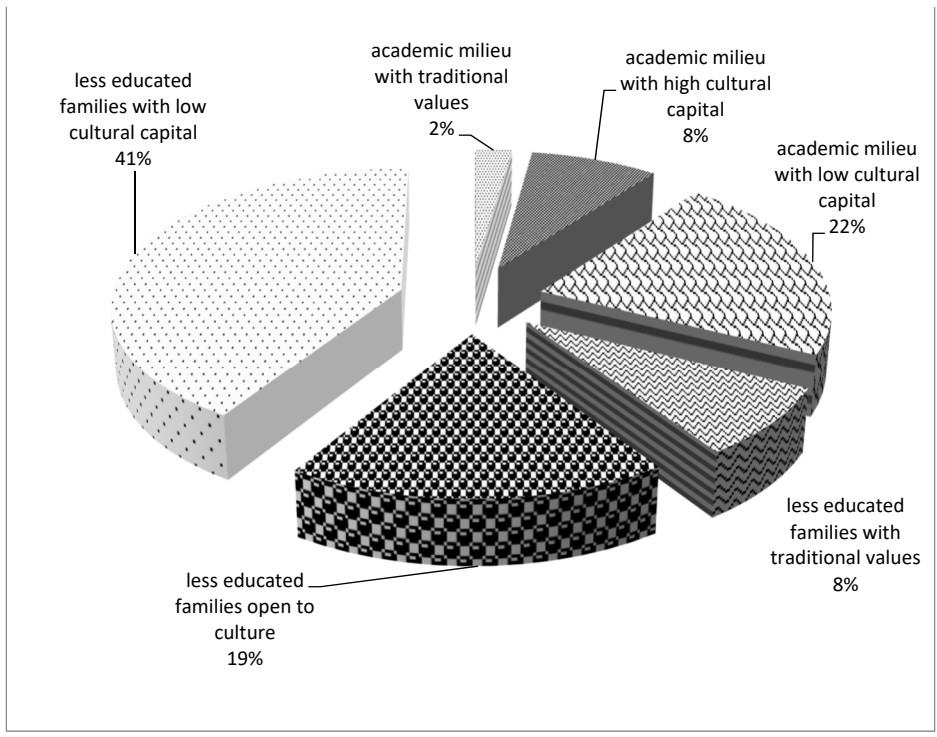

**Figure 2.** The family milieu clusters of the control group. (Source: "Learning music in Hungary 2017" study, composed by author).

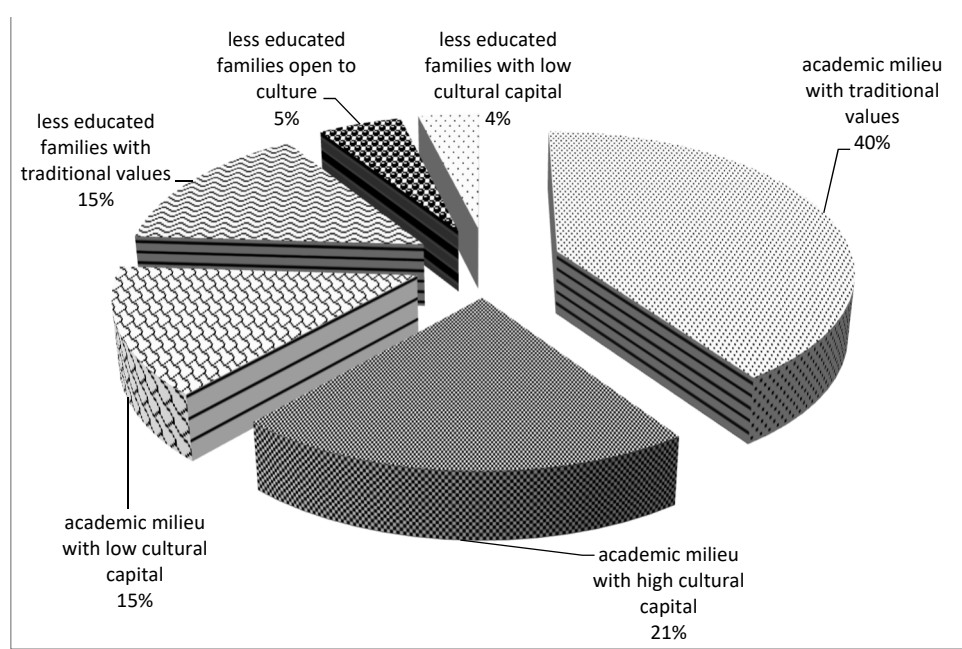

**Figure 3.** The family milieu clusters of the music learner group. (Source: "Learning music in Hungary 2017" study, composed by author).

## 5. Discussion

The main goal of our study was to map out the socio-economic backgrounds of music- and non-music students, and to determine their similarities and differences. Meanwhile, we examined what sort of influential factors exerted effect on participation in music studies. We sought an answer to the question: is music education available expressly for the elite, or does it also have a function in compensating for disadvantage?

We examined the effect of family environment on starting music studies with logistic regression (see Table 1). From among the variables, the objective and relative financial standings, religion, family structure, the mother's diploma, the size of the family library, singing at home and culture consumption habits all displayed influence on the start of music studies. Our familial economic capital analysis results indicate that the more favourable objective financial situation of the family more than doubles the chance of the child's music participation. Contrastingly, the relative financial situation decreases the chance of becoming involved in music. When comparing the objective financial situation to culture consuming habits we did not observe clear-cut correlations between financial situation, cultural interest or participation. Examinations of the objective and relative financial situation of the two groups show differences for the benefit of music students, however, the difference is not significant. Furthermore, the music learner group showed greater extremes both positively and negatively. Relative financial standing, however, decreases the chances of entrance. It is worth noting here that children find it hard to judge their families' financial situations as they consider their everyday circumstances natural. They have little insight into the monetary standing of the family, thus, making a comparison difficult. However, in the family control group we experienced that in terms of real estate they are at a disadvantage compared to the families of music students, whilst being ranked higher for possessing movables of great value. Thus, students in this control group may rightly feel that they live better than their peers. Earlier studies had also found ties between families' monetary well-being and the investments in children's development. These, likewise, point to the unclear relationship (Kaushal et al. 2011; Chin and Phillips 2004).

When analysing social capital, the family structure, the number of siblings and religiosity were taken into account. The results indicate that both being brought up in a traditional family and having a religious family upbringing more than double the chances of the child to join the students in the music learner group. Both variables belong to the

components of traditional values, which are closely related to recognizing the importance of learning music (Pusztai 2009). Furthermore, logistically it is easier for two parents to take their children to elementary art schools than for a single parent. The number of siblings (having at least two siblings) slightly increases the chances of learning music, here, personal experience and the experiences of the siblings positively influence that prospect. Kovács (2016) on sports, as an extracurricular activity, also experienced the positive effect of the already-active-athletic older brothers on the young brother's entrance into sporting.

Using Bourdieu's typology, we grouped the components of cultural capital into three categories: institutionalized capital (e.g., qualifications), objectified cultural capital (e.g., books, devices, paintings), incorporated, internalized cultural capital (e.g., in the form of long-lasting skills of the individual). In the case of the first group, we studied the parents' highest level of education. The results indicate that having parents with higher education degrees increases the chances, especially in the case of the mother's degree, where the chances for the child to enrol in music studies are 3.48 times higher. The highest level of education of the parents affects the extent of time, energy and money that they invest. Parents with a lower level of education are less likely to take risks or put preference on the future, instead, they consider the instantly realized present results more important. That is why they consider investing in the education and the future of their children less important (Boudon 1998; Engler 2010). On the other hand, parents of higher social status dare to take risks and, in the hope of prosperity in the future, they invest in the activities promoting the development of their children, (Mollenhauer 1974; Becker 1998; Chin and Phillips 2004) for instance, in the area of music education, art or sports activities.

In the second category, we studied the size of the family library. Having more than 20 shelves of books increases the chances by more than fourfold. The third group includes singing or playing music together in the family and culture consuming habits. Playing music together shows less influence but singing at home more than doubles the chances of becoming involved in music training. Culture consuming habits showed the greatest impact, which increases the chances more than tenfold. In terms of consuming culture, we examined the interest in high culture. Overall, all three types of capital seem to be influencing the start of music participation, but the most determinate is the effect of cultural capital. In Reeves' (2015) study, he refers to intra-class differences between parents, though he could not find a clear explanation of the latter. In one of the parent groups, conscious use of strategy was at play, behind which the intent to be socially mobile is observable. In the other group, disposition and training are in focus regarding cultural decisions.

During the course of cluster analysis to seek out music- and non-music student attributes, six different types came to light: 1. academic milieu with traditional values; 2. academic milieu with high cultural capital; 3. academic milieu with low cultural capital; 4. less educated families with traditional values; 5. less educated families open to culture; 6. less educated families with low cultural capital, the attributes of which are described below.

The percentage of children belonging to the academic milieu traditional values group stood at 23.1%. While they can be characterized by a favourable financial situation, in terms of possessed assets, a different set of values can be detected among them that are absent in the other groups. Three-quarters of the families own houses, 40% possess flats, and some do not have any estate at all. Three-quarters of them also possess all the items necessary for the comfort and convenience of the family (e.g., car, washing machine). However, plasma TVs, LCD TVs and tablets appear in a smaller proportion (55–68%). This reflects that their set of values is different from that of the other groups, their investment strategy is different. Almost all families (99%) have musical instruments at home. Both parents have higher education degrees, and there are at least three children in the family. They are religious, they regularly sing or play music together at home and they also take part in cultural events.

The academic milieu with high cultural capital group was made up of 15.2% of children. Their supply of material assets is outstanding. Three-quarters of the families own flats, one half possess houses, a quarter of them also possess holiday homes or allotment gardens. Four-fifths of them possess movables of great value. Most of the families (89%)

have own cars and musical instruments (91%). The parents have higher education degrees and they have at least three children. They regularly take part in events of high culture. They regularly sing together at home but do not play music together. They do not consider themselves religious.

The academic milieu with low cultural capital group was made up of 17.9 of students; their financial situation is excellent. Two-thirds of the families possess houses, half of them live in flats, a fifth also have holiday homes or allotment gardens. More than 90% own cars, and more than two-thirds possess movables of great value (such as plasma TVs, LCD TVs, dishwashers). More than half of the families own musical instruments. There are one or two children in these families. The mother has a degree, but the father does not have higher education qualifications. It is worth revisiting the results of the logistic regression, which indicate that the chances of starting music studies increase 3.5 times if the mother is highly educated. The children's cultural interest is enough only for cinema visits or perhaps theatre or library visits about every three months. The children identify themselves as religious. The parents are not actively involved in the music education of the children at home, and they do not sing or play music together.

From less educated families with traditional values 11.8% of children descend. These families are well supplied with material assets. Half of the families live in flats, two-thirds live in houses, but some families among them do not possess any real estate. There are fewer plasma TVs and LCD TVs in this group as well than the intellectual cluster with traditional values. Four-fifths of the families own cars and approximately 90% of the children have smart phones and computers. Three-quarters of the families possess musical instruments. The parents are not highly educated, and they bring up at least three children. They are religious, regularly sing together, but do not play music together. They frequently participate in cultural events.

The number of children of the less educated families open to culture reaches 11.3%. Almost all families (98%) live in flats, only a small proportion owns houses. Four-fifths of them possess movables of great value, however, less than half of the families have musical instruments. The parents do not have higher education qualifications and they belong to the group of large families. They are religious, regularly sing together at home, but rarely attend cultural events.

The less educated families with low cultural capital produced 20.7% of students. They are more modestly supplied with material assets than the previous groups. Two-thirds of the families own flats, 35% possess houses, but 9% do not have any real estate, however, they are well supplied with movables of great value. Three-quarters own cars. About one-third of the families possess musical instruments. The parents do not have higher education degrees, there are one or two children in the families. They are not religious, and they do not attend events of high culture. The parents do not sing or play music together with their children at home.

When they appeared, the families with higher and lower educational qualifications had similar groups, as can be seen in their names. A further interesting point is the fact that the families with traditional values appeared where music education was of emphasized value (Pusztai 2004). Besides this, the open to culture category appeared among the groups with lower educational attainment, which indicates the handicap-compensating power of music education, along with the cultural mobility of families with less socio-economic background (Dumais 2005a).

When characterizing the social groups of the families, our key objective was to observe the proportion of each type of family milieu in the music learner group and the control group. Though the cluster ratio seems to have evened out across the entire sample (see Figure 1), if we examine music students and the control group separately, significant differences surface (see Figures 2 and 3).

The two figures clearly show the differences in the social background of the groups. Less educated families with low cultural capital are dominant in the control group. Here another two major components emerge; the academic milieu with low cultural capital

and the less educated families open to culture. Groups with traditional values appear at a minimum rate. As far as their financial situation is concerned, the housing conditions of the families are less favourable than those of the music students, they more dominantly own flats than houses. Nonetheless, they are well equipped with movables of great value, and they put great emphasis on the exterior, obtaining the fashionable, more spectacular items, which, they expect, will reinforce their social status. Opening up to high culture appears in the group, but two-thirds of the group have low cultural capital. They frequent cinemas, occasionally visit theatres and libraries, but they do not attend any events of high culture (e.g., classical music concerts, folk music events, exhibitions, etc.) About half of the group practise religious activities. Overall, the majority of the students in the control group are children of parents without higher education qualifications who are well supplied with material assets, especially with movables of great value, however, at home, they receive little cultural support from their parents.

The highest proportion of music students were brought up in more highly educated families with traditional values. Moreover, academic milieu with high cultural capital is also dominant. In these families the parents are actively involved in the music education of their children, they sing and play music together, thus, they spend quality pastime together. They possess large family libraries; the transmission of the cultural capital is important to them. The parents' value preferences match with those of the elementary art schools, this way they support the music learning ambitions of their children. Most of the children regularly attend events of high culture. More than four-fifths of the families live a religious way of life. Their supply of material assets is favourable; the majority of these families live in houses. They possess the movables necessary for their studies and comfort, but they are more modest when it comes to having exterior devices. We can conclude that music students arrive at the elementary art schools from a traditional, mainly highly qualified, religious family milieu dominated by cultural capital. The possession of material assets also reflects the traditional set of values, which can be seen in the investing strategy of the families not only in the case of financial matters but also in their investment in culture, schooling, and extracurricular activities.

So, our hypothesis was justified, according to which we assumed that music students are children of families with higher cultural capital, mostly with music education backgrounds, who consider different extracurricular activities investments, which support their advancement in the social hierarchy or help them retain their present status. Yet it is also observable in our study that, also in the case of the better-educated families, the groups with low levels of cultural capital appear. Furthermore, we also find those with much cultural capital among the less educated families. It is worth referring to the dilemma in the special literature, that says differences in the classes are detectable, while the appearance of social reproduction is still unclear. The choice of values by parents, along with "child capital" largely affect the benefits invested in the children's future (Chin and Phillips 2004, p. 187).

The empirical studies have highlighted that both the elitist and the disadvantage compensation effect can be observed in elementary art schools. For the better-educated multigenerational families, learning music can serve as a way of transmitting cultural capital (Andor 2002). In our study, we found that learning music is linked to a particular way of thinking, which gives preference to traditional values (Pusztai 2009). Among the parents of music students, the rate of degree-educated parents was much higher than expected, which confirms the intention of more highly educated families to transmit culture. In the case of highly educated families, a deliberate choice of values can be detected. The higher qualified the parents are, the more likely they are to spend on the accumulation of the child's cultural capital rather than on consumer goods (Andor 2002). However, families who are less educated and with lower cultural capital also appeared among music learners. Thus, music education has become socially more open in elementary art schools and children with lower socioeconomic status have also appeared. Music participation can provide social mobility for the children of lower-status families (DiMaggio 1982; Dumais

2005a). In this way, both the elitist and the disadvantage compensation function can be observed in the elementary art schools with the emphasis varying by region.

The restrictions of the research mean that we examined the long-standing institutions of county seats. To see the whole picture, it is necessary to do research on every settle type, after all, smaller settlements and villages have different infrastructural and personal attributes which affect the study's results. It is in these places that we are likely to see the handicap-compensating role dominate.

The fact that this study details the factors influencing the start of music education and the attributes of children studying music can serve as a good foundation and launchpad for the researchers who deal with this topic in the future. With the broad description of the examined settlement types, as well as the more colorful exploration of parent and child motivations, we can obtain a more complex picture of the causation ties that weigh upon music and art education.

## 6. Conclusions

The most significant results of our research: recognition of the factors affecting the start of music education, as well as the exploration of the attributes of children studying and not studying music, and the characteristics of family milieu.

Though there was no clear connection between families' financial situations and study of art, the better monetary standing of children studying music was still observable (beside many other influential factors). Aside from this, an emerging group coming from among the less educated families was observable. Thus, it is worthwhile to, first, inform parents of the study opportunities and transfer effects of music and art. Second, it is necessary to create as many free, "available goods" courses and occasions, where children can catch the passion for the arts and can try out their own skills. This practice of the arts could be a tool for families of various socio-economic backgrounds and with various amounts of cultural capital, to help their children start off from a more even playing field throughout the course of their school careers.

Along with the fact that culture consumption habits play a significant role in the starting and completion of music education, it is important to mention how critical the shared activities of parents with their children can be in this arena. In the fast-paced world of the 21st century few families devote time and effort to activities or time spent together with their children at home, either by singing or playing music or doing some crafts together (quality time[6]). However, this common activity could build a closer relationship between the two parties, as well as foster a relaxing atmosphere and a joyful experience. This strong bond strengthens trust and reassures children that they can count on their parents if they turn to them with any problem. During adolescence, this is especially critical, as it can protect the youngsters from a lot of unpleasant experiences and arguments (Pribesh et al. 2020). Art activities could be a great help in this, since it provides opportunity to draw in parents in multiple ways as well (Szűcs et al. 2022).

The clusters that developed among the groups of music students and non-music students helped in that we can define in a more concentrated fashion the similarities and differences that exist between the two groups. Among these, it is worth mentioning that the class differences described by Bourdieu are still detectable today, and yet a group emerges from among the less educated families. This occurs where making conscious value choices and taking advantage of opportunities, built upon the talent and strengths of the children, they started out on the road of social ascension. For this, a great tool is music and the study of the arts.

**Funding:** This research was funded by the Scientific Foundations of Education Research Program of the Hungarian Academy of Sciences. And The APC was funded by university of Debrecen.

**Institutional Review Board Statement:** The study was conducted in accordance with the Declaration of Helsinki, and approved by the Institutional Review Board (or Ethics Committee) of University of Debrecen (protocol code 4/2017 and date of approval 25 April 2017).

**Informed Consent Statement:** Informed consent was obtained from all subjects involved in the study.

**Data Availability Statement:** The data presented in this study are available on request from the corresponding author. The data are not publicly available due to the survey group consisting of minors, thus it was the institutions' specific request not to make the data publicly accessible.

**Conflicts of Interest:** The author declares no conflict of interest.

## Notes

1   In Hungary, in elementary art education emphasis is placed on skill and personality development. The teaching material is a tool to improve students' intellectual, emotional, and expressional skills. Improvement and knowledge enrichment is viewed as an instrument for personality formation. The task of elementary art schools is to maintain and development art skills, talents, and, if need be, to prepare students for profession-oriented further study. The training is available to any child who passes the basic audio and rythm drills (drills in the case of music art that require no prior preparation) at the entrance exam. Study takes place at elementary and further study class levels, for ages between 6 and 22. The interested children can choose from four branches of art–music, fine- and industrial arts, puppet- and performing arts, dance art. Maximum six hours a week are set aside for practicing the main subjects and acquiring the theoretical knowledge connected to them. Elementary art school is not obligatory, and students can only take advantage of services upon paying a fee. Students can only student free of charge at these institutes if they are cumulatively disadvantaged, simply disadvantaged, physically, sensory, intellectually handicapped and autistic (The CXC Act of 2011).

2   ISCED (International Standard Classification of Education) 1: elementary school, lower grade (1–4. osztály), ISCED 2 elementary school, upper grade (5–8. osztály), ISCED 3 high school (Forgács 2009).

3   Hungarian National Assessment of Basic Competences: "Since 2008 every student in the 6th, 8th and 10th years of their school education has had to complete competency test sheets and a background questionnaire revealing their socioeconomic status and attitudes towards learning. The assessment is similar to PISA in several respects, but it is limited to the assessment of reading comprehension and mathematics." (Bacskai and Pándy 2017, p. 210).

4   To measure relative financial status, the respondent had to mark on a 10-point scale where they would place their family compared to other families. Based on the distribution of relative financial status and its peaks, we created three groups – below average (1–5), average (6), and above average (7–10)—which provided opportunity for further comparison. It is most definitely worth noting in connection iwth this variable that, on the one hand children struggle to judge their family's monetary situation. This is a natural state to them, since they live in it daily, and do not see into the financial positions of other families—this makes the comparison a challenge.

5   During the examination of cultural consumption habits, we asked about the frequency of theater, cinema, museum, exhibition, classical music concert, folk music event, and library visitations. The "high culture consumers" library, theater, and music concerts are visited monthly, while the other events are frequented every three months.

6   David H. Demo (1992): Parent-Child Quality Time: Does Birth Order Matter?

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
