# Peer review of "The Relationship between Family Milieu and Music Education"

_socsci, doi:10.3390/socsci11120579_

Round 1

Reviewer 1 Report

Please see the document attached. 

Author Response

Thank you for the constructive suggestions that helped me to improve the quality of the study.
I completely revised the study, I would like to attach it if possible.

Reviewer 2 Report

Dear Authors, 

First of all, I would like to congratulate you on your work, since you present an interesting topic, approached from Bordieu's point of view. Although the work is of great value, my assessment is that it needs major modifications in order to be considered for publication, so my recommendations are as follows: 

- The article studies a control population and the sample of art students, as they state on several occasions. E.g. " We sampled eight-grade students of elementary art 6 schools in several county seats that had a long history of teaching (n=270) and eighth-grade students 7 in several elementary schools (n=285) as a control group. We assumed that students learning music 8 are children of families with higher cultural capital, mostly with backgrounds in music education, 9 who consider extracurricular activities investments.", throughout the article they explain what led them to make the decision, also making reference to the fact that the control population belongs to the middle class, although they do not explain in depth how this may affect the student, and to what extent this control population can be compared with students in art schools in terms of capital. I recommend a more in-depth explanation of this aspect. 

- In addition to this, the introduction is too brief and does not go into the background, and the methodology begins with a section that could be considered part of the theoretical framework, I recommend clarifying these sections, delimiting them according to the information presented. 

- As for the description of the analysis, it would be interesting to provide information on how the process was carried out, beyond stating that the methods included logistic regression and cluster analysis.

- Finally, it would also be advisable to separate the sections on results and discussion, in this way it would be easier to identify what is the relevance and contribution of the study, it is important to reinforce this idea in a clear way. 

Best wishes, 

Reviewer.

Author Response

I really appreciate your suggestions and detailed guidance. I feel that my study has become much better and more understandable thanks to the corrections.

I would like to attach the corrected study.

Round 2

Reviewer 1 Report

Amazing improvement. I really enjoyed reading it, and it was easy to follow and understand. I had previously expressed that your topic was fascinating, so congrats!

Few minor edits:

Found a typo in line 52, it should be "conclusions."

Lines 55-56 and 60-61 seem to repeat the same idea. I recommend deleting one of them.

Lines 74-75, I recommend removing the word"things" and finding a better way to express that Bourdie and Coleman have different approaches or views on the same topic

Line 606, if this is APA style and "child capital" is a direct quote, I think the page number is missing.

Author Response

Thank you for the nice words. I corrected everything you suggested and fixed the references as well.

Reviewer 2 Report

I feel that my concerns have been met. And after a revision of references and typos, the article could be accepted. 

Author Response

Thanks again for your help and guidance. I corrected the typos and put the references in order.